# Global Stability of the Curzon-Ahlborn Engine with a Working Substance That Satisfies the van der Waals Equation of State

**DOI:** 10.3390/e24111655

**Published:** 2022-11-14

**Authors:** Juan Carlos Pacheco-Paez, Juan Carlos Chimal-Eguía, Ricardo Páez-Hernández, Delfino Ladino-Luna

**Affiliations:** 1Basic Sciences Departament, Metropolitan Autonomus University, Campus Azcapotzalco, Av. San Pablo 180, Col. Reynosa, Mexico City CP 02200, Mexico; 2Laboratorio de Simulación y Modelado, Centro de Investigación en Computación, Instituto Politécnico Nacional, Av. Juan de Dios Bátiz s/n UP Zacatenco, Mexico City CP 07738, Mexico; 3Área de Física de Procesos Irreversibles, Basic Sciences Departament, Metropolitan Autonomus University, Campus Azcapotzalco, Av. San Pablo 180, Col. Reynosa, Mexico City CP 02200, Mexico

**Keywords:** Curzon–Ahlborn, Van der Waals, nonlinear dynamic, Lyapunov, global stability, time delay

## Abstract

In this paper, we show an analysis of the global stability of a Curzon–Ahlborn engine considering that the working substance of the engine satisfies the Van der Waals equation of state, which is more general than the ideal gas case. We use the Lyapunov stability theory for the case where the engine operates at a maximum power output. We analyze the steady state of the intermediate temperatures as well as the asymptotic behavior of the performance of the engine. Additionally, we study the relationship between the inherent time delay by analyzing the dynamic properties of the system and the stability of the steady state. We present illustrative graphs of the obtained results. Finally, we include a brief discussion of the obtained results and appropriate conclusions.

## 1. Introduction

Since the Curzon–Ahlborn paper, published in 1975 [1], the so-called finite-time thermodynamics (FTT) started its development as a branch of classical equilibrium thermodynamics. Diverse ways and methods to qualify the performance of the Curzon and Ahlborn engines and other heat engines have been obtained [2,3,4,5,6,7]. One of the main objectives of FTT is the analysis of the optimal operation of thermal engines to obtain new operating limits of efficiency and power output in thermal devices. That is, there was no thermal equilibrium between the working fluid and the thermal reservoirs in the isothermal branches of the cycle. Curzon and Ahlborn (CA) showed that such a heat engine provides nonzero power, positive entropy production, and a more realistic efficiency than the Carnot engine. In Figure 1, we show the representation of a CA engine, in which $T_{1}$ and $T_{2}$ are the temperature of the heat reservoirs with $T_{1}>T_{2}$, $J_{1}$ and $J_{2}$ are the heat fluxes of the engine through thermal conductors, both with the same conductance (α) and the same heat capacity (C), and x and y are the hot and cold temperatures of the Carnot cycle isothermal branches, respectively.

The efficiency of the CA engine (Figure 1) at maximum power output is given by $\eta_{CA}=1-\sqrt{\tau }$, (with $\tau =T_{2}/T_{1}$ and $T_{1}>T_{2}$). This efficiency is a better approximation for real heat engines than the one proposed by Carnot. Most studies in FTT of thermal systems have focused on their steady-state energy properties, which are important from a design viewpoint. FTT studies of the dynamic properties [3,4,7] of the system have also been conducted, such as the response to a noisy disturbance or the stability of the steady state of the system. Recently, studies have proposed to analyze the stability and robust dynamics of thermal engine models. In real systems, when modifying the value of a variable, the change in the system is not appreciated immediately, but a certain amount of time must lapse to observe the effect of said change in the dynamic response of the system. These systems are known as time dynamic systems. The robustness of a system can manifest itself in one of two different ways: the system returns to its current dynamic attractor after the disturbance, or it moves to a different attractor that maintains the function of the system.

Recently, the distinct types of stability problems in dynamical systems have been studied through stability theory, which plays a significant role in systems and engineering studies. In 2001, Santillan et al. [8] conducted the first work on the local stability analysis of a CA endoreversible engine working at maximum power output. These authors showed that a clever design generates stability between dynamic robustness and optimal thermodynamic properties. Páez-Hernández et al. [9,10] studied the effects of robustness and time delay on the performance and local stability properties of a non-endoreversible Curzon–Ahlborn engine. Páez-Hernández et al. [11] studied dynamic properties in an endoreversible Curzon–Ahlborn engine using a Van der Waals gas as a working substance. Páez-Hernández et al. [12] studied the effects of time delays in endoreversible and non-endoreversible thermal engines working at different regimes. Chimal-Eguía J. C. et al. [13] analyzed the stability of a Curzon–Ahlborn engine with the Stefan–Boltzmann heat transfer law. Guzman-Vargas et al. [14] performed local stability analysis of the CA engine under different heat transfer laws. Barranco-Jiménez M. A. et al. [15] analyzed the local stability of a thermo-economic model of a Novikov–Curzon–Ahlborn heat engine. Other authors have studied the stability under different operating regimes, such as the regime of maximum efficiency and maximum ecological function [16,17,18,19,20,21,22,23,24]. It is important to note that the finite time thermodynamics continues to address more systems of current interest, as shown in [25,26,27,28,29,30].

Part of the stability theory is the equilibrium points that are generally analyzed by Lyapunov’s stability theorem. An equilibrium point is said to be stable if all solutions that start in the vicinity of the equilibrium point remain in the vicinity of the equilibrium point; otherwise, the equilibrium point is unstable. If a stable point fulfills that all the solutions that start in the vicinity of the stable point remain in the vicinity of the equilibrium point, and also tends towards equilibrium as time approaches infinity, then the equilibrium point is said to be asymptotically stable.

Different models of thermal engines can be studied [31,32,33,34] using the Lyapunov stability theory for the analysis of the dynamic properties of the system. In these studies, it is assumed that the working substance of the engine is an ideal gas. However, in real engines, this substance is a non-ideal gas. In this paper, we analyze the influence of assuming a non-ideal gas as the working substance in the engine, particularly a Van der Waals gas. We note that the efficiency of the CA engine, at maximum power output, can be written as the limit at zero of a series of powers of a factor involving the compression ratio, as shown by Gutkowicz-Krusin et al. [35]. In the case of a Van der Waals gas, this efficiency has the same form, and only differs in that the noted factor includes the available volume of one mole of particles; Ladino-Luna D [36]. We use the Lyapunov stability theory and show the global stability analysis of a Curzon–Ahlborn engine with a working substance that obeys the Van der Waals equation of state when the engine operates at maximum power output. We analyze the steady state of the intermediate temperatures, as well as the delay times of the machine. The work is organized as follows. In Section 2, we present the steady-state characteristics of the Curzon and Ahlborn engine with a Van der Waals gas (CAEVW). The stability analysis and effects of the time delay of the CAEVW are presented in Section 3. The global stability of the CAEVW is presented in Section 4. For each important result, we show illustrative graphs of the obtained results. Finally, in Section 5, important conclusions and remarks are presented.

## 2. Steady-State Characteristics of the CAEVW

The heat engine, shown in Figure 1, operates between temperatures $T_{1}$ and $T_{2}$, with $T_{1}>T_{2}$. We can consider the steady-state temperatures as $\bar{x}$ and $\bar{y}$, where $T_{1}>\bar{x}>\bar{y}>T_{2}$. Considering a Newton heat transfer law, where the heat exchange is carried out through the thermal conductors, both with conductance α, the irreversible fluxes from $T_{1}$ to $\bar{x}$ and from $\bar{y}$ to $T_{2}$ are ${\bar{J}}_{1}$ and ${\bar{J}}_{2}$, respectively, where ${\bar{J}}_{1}$ and ${\bar{J}}_{2}$ represent the steady-state heat fluxes of the engine,
(1)$${\bar{J}}_{1}=\alpha \left(T_{1}-\bar{x}\right)$$
and
(2)$${\bar{J}}_{2}=\alpha \left(\bar{y}-T_{2}\right)$$

The following inequality follows from the Clausius theorem and the fact that the inner Carnot-like engine works in irreversible cycles:$$\frac{{\bar{J}}_{1}}{\bar{x}}-\frac{{\bar{J}}_{2}}{\bar{y}}<0$$

Considering the endoreversibility hypothesis, the above equation results in
(3)$$\frac{{\bar{J}}_{1}}{\bar{x}}=\frac{{\bar{J}}_{2}}{\bar{y}}$$

The system’s steady-state power output and efficiency can be defined as
(4)$$\bar{P}={\bar{J}}_{1}-{\bar{J}}_{2}$$
and
(5)$$\bar{\eta }=\frac{\bar{P}}{{\bar{J}}_{1}}=1-\frac{{\bar{J}}_{2}}{{\bar{J}}_{1}}\;$$

By combining Equations (1)–(3), and (5), we can solve for $\bar{x}$ and $\bar{y}$ in terms of *T*_1_, *T*_2_ and $\bar{\eta }$ as
(6)$$\bar{x}=\frac{T_{1}}{R+1}\left(1+\frac{\tau }{1-\bar{\eta }}\right)\;$$
(7)$$\bar{y}=\frac{RT_{1}}{R+1}\left(1-\bar{\eta }\right)\left(1+\frac{\tau }{1-\bar{\eta }}\right)$$
where $\tau =T_{2}/T_{1}$.

In our analysis, we use a working substance that obeys the Van der Waals equation of state, which, unlike the ideal gas equation PV=nRT, considers a correction in the volume of the molecules, which are treated as small hard spheres, leading to a minimum volume b (for 1 mol). That is, in the ideal gas equation V is replaced by V−b. Additionally, the Van der Waals equation considers the attraction between molecules, which tends to decrease the volume, considering a small superficial layer of gas; its attraction for the internal molecules should be proportional to the square of the density.

Thus, the efficiency of the Curzon–Ahlborn heat engine, operating at maximum power output and considering a Van der Waals gas working substance following [36], is given by
(8)$$\eta_{vW}=1-\left\{\sqrt{\tau }+\frac{1}{2}{\left(1-\sqrt{\tau }\right)}^{2}\lambda_{vW}+\frac{1}{4}{\left(1-\sqrt{\tau }\right)}^{2}\left[\frac{{\left(1-\sqrt{\tau }\right)}^{2}}{2\sqrt{\tau }}-ln\tau \right]\lambda_{vW}{}^{2}+O\left(\lambda_{vW}^{3}\right)\right\}$$

With
(9)$$\lambda_{vW}=\frac{1}{\gamma -1}{\left(ln\frac{V_{max}-b}{V_{min}-b}\right)}^{-1},$$
where γ is the ratio of the heat capacities, $V_{max}$ and $V_{min}$ are the volumes supplied by the gas in one cycle, and *b* is a gas-dependent constant. Nevertheless, the *b* parameter in the Van der Waals equation is so small compared with the spanned volumes of the engine, namely $V_{max}$ and $V_{min}$. Therefore, we have $\frac{V_{max}-b}{V_{min}-b}≅\;\frac{V_{max}}{V_{min}}$, and $\frac{V_{max}}{V_{min}}=r_{c}$ is the compression ratio of the engine, which has a particular value depending on the chosen cycle: Diesel, Otto, etc. In the case of the Diesel cycle, $12\leq r_{c}\leq 15$; for the Otto cycle, $2\leq r_{c}\leq 15$ [37]. Considering $r_{c}=12$ and substituting in Equation (9), we obtain $\lambda_{vw}=1.006$, for the simplicity calculus we use $\lambda_{vw}=1$, now substituting this value in (8), we obtain the efficiency $\eta_{VW}=\eta_{C}/2$. This substitution of lambda is justified by observing the behavior of the plots in Figure 2. It is even shown an interval of τ, namely $\tau \;\epsilon \left[0.322,0.674\right]$, in which real plants usually work [38,39]. As we can see, the behavior of $\eta_{VW}$ and $\eta_{C}/2$ is practically the same in that interval; so there is no loss of generality in the present study if we suppose $\lambda_{vW}=1$, and we obtain
(10)$$\eta_{vW}=1-\sqrt{\tau }-\frac{1}{2}{\left(1-\sqrt{\tau }\right)}^{2}=\frac{1-\tau }{2}=\frac{\eta_{C}}{2},$$
where $\eta_{C}$ represents the Carnot efficiency. Substituting Equations (6) and (7) into Equation (10), to express the steady state to the CAEVW and the results
(11)$$\bar{x}=\frac{T_{1}}{2}\left(\frac{1+3\tau }{1+\tau }\right)$$
and
(12)$$\bar{y}=\frac{T_{1}}{4}\left(1+3\tau \right).$$

For the power output of the CAEVW in the steady state, by combining Equations (1), (5) and (6) into Equation (10), we obtain
(13)$$\bar{P}=\frac{T_{1}}{4}\frac{{\left(1-\sqrt{\tau }\right)}^{2}}{\left(1+\tau \right)}.$$

To perform the stability analysis, we write $T_{1}$, $T_{2}$, and $\bar{P}$ in terms of $\bar{x}$ and $\bar{y}$, that is
(14)$$T_{1}=-\frac{2\bar{x}\bar{y}}{\bar{x}-3\bar{y}}$$
(15)$$T_{2}=\frac{2}{3}\left(2\bar{y}+\frac{2\bar{x}\bar{y}}{\bar{x}-3\bar{y}}\right)$$
(16)$$\bar{P}=-\frac{\alpha \left(\bar{x}-\bar{y}\right)}{\bar{x}-3\bar{y}}$$

## 3. Stability Analysis and Effects of Time Delays of the CAEVW 

### 3.1. Dynamic Model of the CA Engine 

As was noted, Figure 1 is the representation of a CA engine, in which $T_{1}$ and $T_{2}$ are heat reservoirs with $T_{1}>T_{2}$; $J_{1}$ and $J_{2}$ are the heat fluxes of the engine through thermal conductors, both with the same conductance (α) and the same heat capacity (C); x and y are the hot and cold temperatures of the Carnot cycle isothermal branches, respectively. From the capacity definition, $C=\frac{dQ}{dT}$, we obtain the dynamics equation, $\frac{dx}{dt}=\frac{1}{C}\frac{dQ}{dt}$. Now, considering the rate change of the internal temperatures x and y, and the thermal conductance *α*, these temperatures can vary with time, according to the following differential equations:(17)$$\frac{dx}{dt}=\frac{1}{C}\left(\alpha \left(T_{1}-x\right)-J_{1}\right)$$
(18)$$\frac{dy}{dt}=\frac{1}{C}\left(J_{2}-\alpha \left(y-T_{2}\right)\right),$$
with $J_{1}$ the heat flow from x to the working substance and $J_{2}$ the flow from the Carnot engine to y. Both equations cancel when x, y, $J_{1}$, and $J_{2}$ take their steady-state values. We write $J_{1}$ and $J_{2}$ in terms of x, y and the output power of the engine,
(19)$$J_{1}=\frac{x}{x-y}P$$
(20)$$J_{2}=\frac{y}{x-y}P.$$

Assuming that the power output of the CA engine is related to the temperatures x and y, as in the case of the power output for steady-state that is related to the temperatures $\bar{x}$ and $\bar{y}$, then we can write the power output of the CA engine out of steady-state in terms of x and y of the form
(21)$$P=\frac{\alpha {\left(x-y\right)}^{2}}{3y-x}.$$

Now, by substituting Equations (19)–(21) into Equations (17) and (18), we obtain the following expressions in terms of the temperatures x, $y,\;\;T_{1}$, and $T_{2}$,
(22)$$\frac{dx}{dt}=\frac{\alpha \left[T_{1}\left(x-3y\right)+2xy\right]}{C\left(x-3y\right)}$$
(23)$$\frac{dy}{dt}=\frac{\alpha \left[T_{2}\left(x-3y\right)+2y\left(x-2y\right)\right]}{C\left(x-3y\right)}.$$

### 3.2. Local Stability Analysis 

To perform a local stability analysis of the CAVEW dynamic model, we transform Equations (22) and (23) as the following set of ordinary differential equations:(24)$$\frac{dx}{dt}=f\left(x,y\right)\;$$
(25)$$\frac{dy}{dt}=g\left(x,y\right).$$

The system’s steady state is couple $\left(\bar{x},\bar{y}\right)$ that simultaneously satisfies $f\left(\bar{x},\bar{y}\right)=0$ and $g\left(\bar{x},\bar{y}\right)=0$. With a little algebra, it is easy to show that the Curzon–Ahlborn engine has a unique steady state, given by Equations (11) and (12). Based on Strogatz [40], the local stability in the steady state is determined by the eigenvalues of the Jacobian matrix:(26)$$J={\left(\begin{array}{cc} f_{x} & f_{y}\\ g_{x} & g_{y} \end{array}\right)}_{\bar{x},\bar{y}}$$
where
(27)$$f_{x}={\left.\frac{\partial f}{\partial x}\right|}_{\bar{x},\bar{y}}=-\frac{6\alpha {\left(1+\tau \right)}^{2}}{C{\left(1+3\tau \right)}^{2}}$$
(28)$$f_{y}={\left.\frac{\partial f}{\partial y}\right|}_{\bar{x},\bar{y}}=\frac{8\alpha }{C{\left(1+3\tau \right)}^{2}}$$
(29)$$g_{x}={\left.\frac{\partial g}{\partial x}\right|}_{\bar{x},\bar{y}}=\frac{2\alpha {\left(1+\tau \right)}^{2}}{C{\left(1+3\tau \right)}^{2}}$$
(30)$$g_{y}={\left.\frac{\partial g}{\partial y}\right|}_{\bar{x},\bar{y}}=-\frac{4\alpha \left(1+\tau \left(2+3\tau \right)\right)}{C{\left(1+3\tau \right)}^{2}}.$$

Following Páez-Hernández et al. [9] step by step, both eigenvalues have negative real parts, which implies that the system’s steady state is a stable node, and allows us to define relaxation times $t_{i}=-1/\lambda_{i}$, i = 1, 2. The time evolution of a given perturbation from the steady state is generally determined by both relaxation times. However, the perturbation long-term behavior is dominated by the longest relaxation time. In Figure 3, the relaxation times $t_{1}$ and $t_{2}$ are plotted against τ. Notice that $t_{1}>t_{2}$ for all values of τ, that $t_{2}$ is a monotone-increasing function of τ, and that $t_{1}$ has a maximum at τ=0 and a minimum at τ=1.

### 3.3. Dynamic Effects of the Time Delay

Consider a small perturbation near the steady state. It is determined by the linear combination based on $\exp\left(\lambda_{i}t\right)$, i = 1, 2, …, being $\lambda_{i}$ solution of the characteristic equation, McDonald [41],
(31)$$\frac{dx}{dt}=\frac{\alpha \left[T_{1}\left(x-3y_{\pi /2}\right)+2xy_{\pi /2}\right]}{C\left(x-3y_{\pi /2}\right)}$$

Therefore, we can consider the delay differential system equations
(32)$$\frac{dy}{dt}=\frac{\alpha \left[T_{2}\left(x_{\pi /2}-3y\right)+2y\left(x_{\pi /2}-2y\right)\right]}{C\left(x_{\pi /2}-3y\right)},$$
where π/2 represents the variable with time delay.

Because the characteristic equation has an infinite number of solutions, the stability analysis of a dynamical system with time delay can be complicated. Furthermore, as it is known, a common effect of time delays is to destabilize previously stable steady states, as it induces sustained oscillations. In Páez-Hernández et al. [12], it is shown that no time delay can destabilize the CAEVW.
(33)$$\left(f_{x}-\lambda \right)\left(g_{y}-\lambda \right)-f_{y}g_{x}\exp\left(-\lambda \pi \right)$$

Figure 4 shows that surfaces $\rho_{1}$ and $\rho_{2}$, which result from solving Equation (33), do not intersect at any point.

## 4. Global Stability of the CAEVW

### 4.1. Lyapunov Function Description

In this section, we apply the Lyapunov stability theory to analyze and check the global asymptotic stability for the CA heat engine operating at maximum output power by constructing the Lyapunov function. This function, denoted by $V\left(x,y\right)$, must satisfy the following conditions:
(1)$V\left(x,y\right)$ must be positive definite in a region around the steady state;(2)$\dot{V}\left(x,y\right)$ must be negative definite, that is, $\dot{V}\left(x,y\right)\leq 0$.


### 4.2. Lyapunov Function for CA Engine Model

To build the matrix $J\left(x\right)$, we consider the right-hand side of the differential equations given by Equations (22) and (23) and after evaluating the steady-state temperatures in terms of τ, i.e., Equations (11) and (12).
(34)$$J=\frac{2\alpha }{C{\left(1+3\tau \right)}^{2}}\left[\begin{array}{cc} -6{\left(1+\tau \right)}^{2} & 5+\tau \left(2+\tau \right)\\ 5+\tau \left(2+\tau \right) & -4\left(1+\tau \left(2+3\tau \right)\right) \end{array}\right]$$

Considering that the previous matrix is negative definite, the Lyapunov function is written as
(35)$$V\left(x,y\right)=\frac{\alpha^{2}}{C^{2}}\left[{\left(\frac{\left(T_{1}\left(x-3y\right)+2xy\right)}{\left(x-3y\right)}\right)}^{2}+{\left(\frac{\left(T_{2}\left(x-3y\right)+2y\left(x-2y\right)\right)}{\left(x-3y\right)}\right)}^{2}\right]$$

It can be verified in the previous equation that in the stationary state $V\left(\bar{x},\bar{y}\right)=0$, if we consider $\bar{x}$ and $\bar{y}$. 

On the other hand, the function fulfills
(36)$$\dot{V}\left(x,y\right)=\frac{\partial V}{\partial x}\dot{x}+\frac{\partial V}{\partial y}\dot{y}=\nabla V,\dot{X}\leq 0$$
where
(37)$$\frac{\partial V}{\partial x}=\frac{4y^{2}\left(-3T_{1}\left(x-3y\right)+T_{2}\left(x-3y\right)+4y\left(-2x+y\right)\right)\alpha^{2}}{C^{2}{\left(x-3y\right)}^{3}}$$
(38)$$\frac{\partial V}{\partial y}=\frac{4\left(\left(T_{1}-T_{2}\right)x^{3}+x^{2}\left(-3T_{1}+7T_{2}+4x\right)y-6x\left(3T_{2}+2x\right)y^{2}+2\left(9T_{2}+14x\right)y^{3}-24y^{4}\right)\alpha^{2}}{C^{2}{\left(x-3y\right)}^{3}}$$

Figure 5 shows the behavior of the level curves of the Lyapunov function built for our dynamic system defined by Equations (22) and (23). The behavior indicates that, as the constant k of the level curve decreases, the surface also decreases towards the steady-state values $\left(\bar{x},\bar{y}\right)$. Our result is consistent with the work reported by Guzmán-Vargas L et al. [33].

In Figure 6, the surface of the Lyapunov function is shown. Comparably to these authors [33], we conduct the numerical study of the decay of the disturbances around the stationary state through Equations (17) and (18), for different initial conditions.

## 5. Results

In the numerical analysis shown in Figure 7, the initial state is $\left(x_{0},y_{0}\right)$, and we consider the following cases: (1) $x_{0}>\bar{x}$, $y_{0}>\bar{y}$; (2) $x_{0}>\bar{x}$, $y_{0}<\bar{y}$; (3) $x_{0}<\bar{x}$, $y_{0}>\bar{y}$; and (4) $x_{0}<\bar{x}$, $y_{0}<\bar{y}$. For all cases analyzed, it is observed that they tend to the value of the steady state of the temperature.

## 6. Conclusion and Remarks

In this work, we performed a global stability analysis of a CA engine using a Van der Waals gas as a working substance and by applying the Lyapunov stability theory. For the global analysis, we first studied the dynamics of the CA engine to shape the Lyapunov function, and then we analyzed the global properties. Our results show that the stability of the engine is maintained around the stable state even when variations in parameters of the Lyapunov function are considered, i.e., the surface also decreases towards the steady-state values $\left(\bar{x},\bar{y}\right)$, as reported by other authors. In addition, we studied the local analysis of a CA engine using a Van der Waals gas as a working substance and have considered the engine’s inherent time delays. Time delays are present in multiple systems subject to dynamic regulation. In the endoreversible and non-endoreversible Curzon–Ahlborn engine, the inherent time delays are incapable of destabilizing the steady state [9,12]. Thus, they seem not to play a role in the trade-off between energetic and dynamic properties. This does not have to be true for all energy-converting systems, however. They should be taken into account in further studies concerning the relationship between the dynamic and the energetic properties of energy-converting systems.

## Figures and Tables

**Figure 1 entropy-24-01655-f001:**
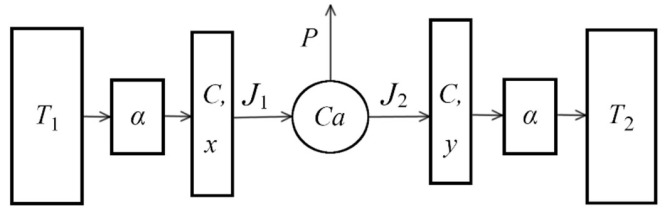
Curzon–Ahlborn engine.

**Figure 2 entropy-24-01655-f002:**
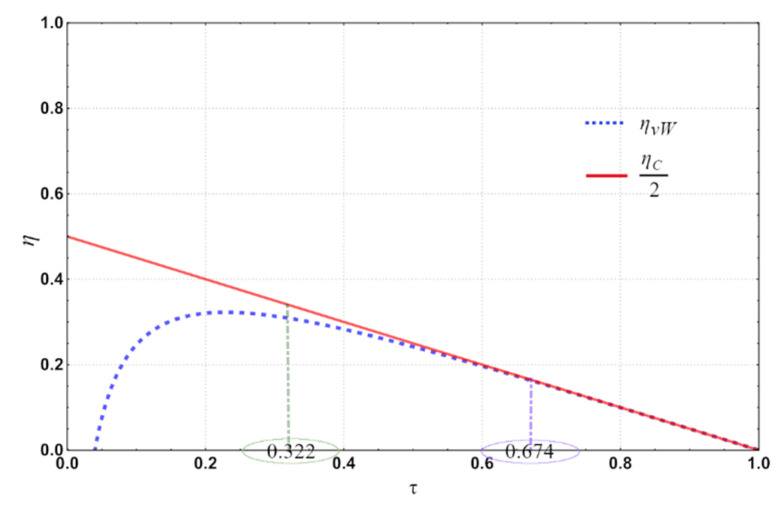
The behavior of $\eta_{VW}$ and $\eta_{C}/2$ in which real plants usually work.

**Figure 3 entropy-24-01655-f003:**
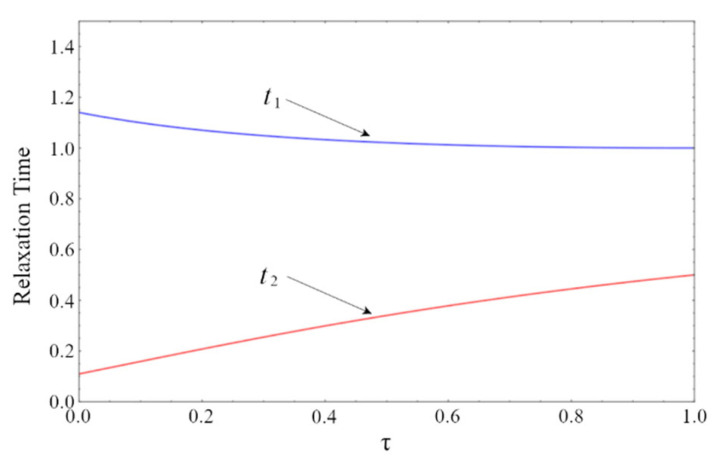
Plots of the relaxation times $t_{1}$ and $t_{1}$ in units of C/α, vs. τ.

**Figure 4 entropy-24-01655-f004:**
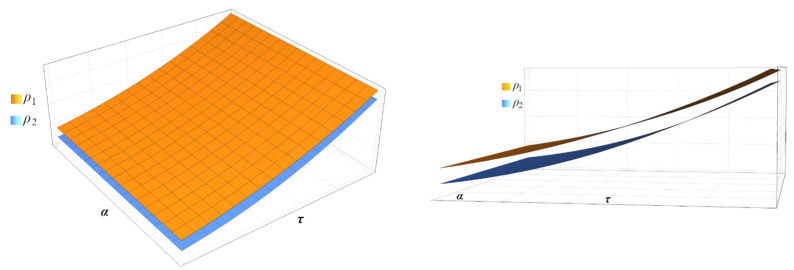
Surface plot of the effects of time delay that shows that CAEVW cannot be destabilized by any time delay.

**Figure 5 entropy-24-01655-f005:**
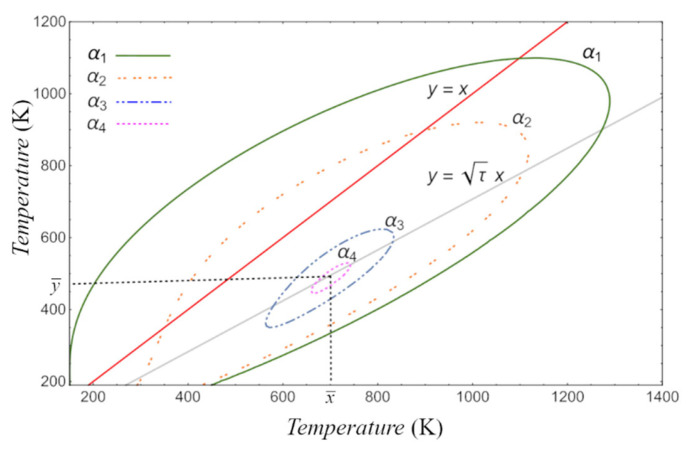
Level curves of the Lyapunov function $V\left(x,y\right)$. The constant on each level curve satisfies $k_{4}<k_{3}<k_{2}<k_{1}$.

**Figure 6 entropy-24-01655-f006:**
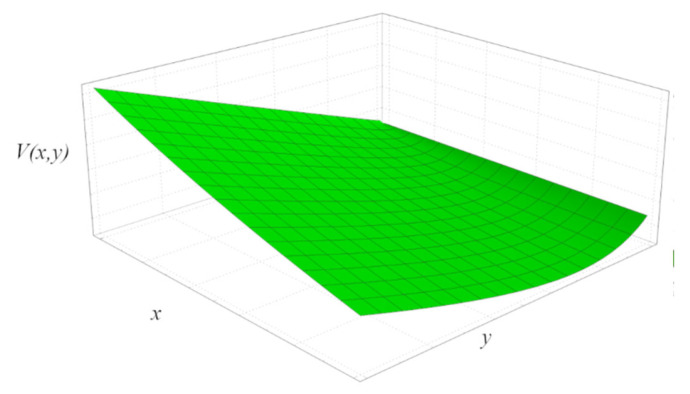
Surface plot of the Lyapunov function.

**Figure 7 entropy-24-01655-f007:**
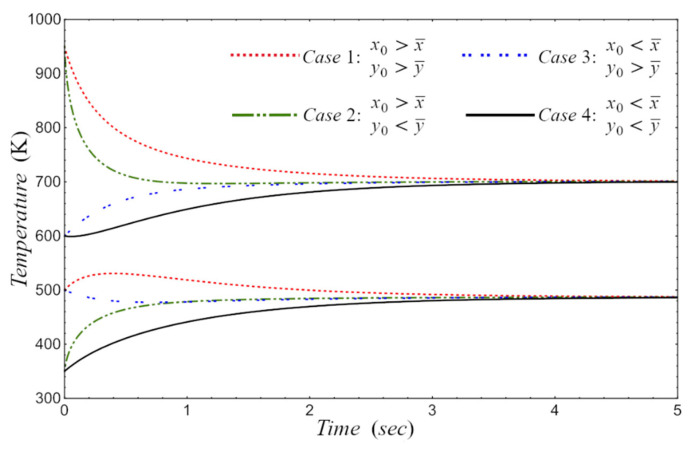
Working temperature vs. time, considering: (1) $x_{0}>\bar{x}$, $y_{0}>\bar{y}$; (2) $x_{0}>\bar{x}$, $y_{0}<\bar{y}$; (3) $x_{0}<\bar{x}$, $y_{0}>\bar{y}$; and (4) $x_{0}<\bar{x}$, $y_{0}<\bar{y}$.

## Data Availability

Not applicable.

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
