# Peer review of "Global Stability of the Curzon-Ahlborn Engine with a Working Substance That Satisfies the van der Waals Equation of State"

_entropy, 2022, doi:10.3390/e24111655_

Round 1

Reviewer 1 Report

I have read the manuscript Global stability of the Curzon-Ahlborn engine with a working substance that satisfy the Van der Waals equation state, by J. C Pacheco-Paez et al. 

There are some comments/questions that must be addressed before I can give my recommendation for publication.

  1. The abstract can be improved to contain more specific results obtained in the paper.
  2. Additionally, the contribution of the paper to the existing literature must be highlighted in the introduction. The general idea is clear but the specific contribution is not given in advance.
  3. From Eq. 9 to 10 the approximation \lambda_{\nu W}->1 is considered, please add a comment on the physical relevance/interpretation of such case.
  4. Eqs. 11 to 16 are working-fluid dependent? Please add a comment on this.
  5. Include a physical interpretation of the dynamical equations given by Eqs. 17 and 18. Although it seems standard, for non-experts the choosing of this specific dynamic is not clear.
  6. Regarding Eq. 8. Since the \lambda is near 1 (for the approximation in Eq. 10), how do the terms of order 3 and beyond are neglected?
  7. I do not see the influence of the equation of the state parameters in the analysis of stability. Since the stability analysis is built from the efficiency it seems that the working fluid information is lost when the parameters of the model are fixed to obtain the \eta_C/2 efficiency. This should be clarified in the paper and in the concluding remarks.

Author Response

The manuscript was substantially modified according to the comments, so it is necessary to mention that the paragraphs do not follow the original order of the manuscript, but all the observations were made. 

  1. The abstract can be improved to contain more specific results obtained in the paper.

We agree. The Abstract was modified according to your observations.

  1. Additionally, the contribution of the paper to the existing literature must be highlighted in the introduction. The general idea is clear, but the specific contribution is not given in advance.

We agree and the Abstract was modified, taking your observations into account.

  1. From Eq. 9 to 10 the approximation \lambda_{\nu W}->1 is considered, please add a comment on the physical relevance/interpretation of such case.

           Using the Gutkowicz-Krusin et al. formalism [35], Ladino-Luna [36]  obtained ec. (8). Nevertheless, b parameter in the van der Waals equation is negligible compared with the spanned volumes of the engine, namely   and . So we have  , and  is the compression ratio of the engine, which has a particular value depending on the chosen cycle: Diesel, Otto, etc. In the case of the Diesel cycle, ; and for the Otto cycle, . Thus, considering  and substituting it in (9)  we obtain ,  for the simplicity calculus, we use  and substitute it in (8) we obtain   This substitution, by observing the behavior of the plots   as functions of , with , leads to the behavior shown in figure 2. It is even shown an interval of , namely, , in which real plants usually work;  Bejan A., 1988, Advanced Engineering Thermodynamics (New York: Wiley) [38], Velasco S., Roco J.M.M., Medina A., White J.A., and Calvo-Hernández A., 2000, J. Phys. D: Appl. Phys. 33 355 [39]. The behavior of  and  is the same, so there is no loss of generality in this study.

  1. 11 to 16 are working-fluid dependent? Please add a comment on this.

When we obtain   , Eqs. (11-16) we provide information on the Van der Waals state equation

  1. Include a physical interpretation of the dynamical equations given by Eqs. 17 and 18. Although it seems standard, for non-experts the choosing of this specific dynamic is not clear.

The calorific capacity is given as ,  C  or    or     where J is the flux, as we can see in Fig 1, there are two flows,  and

  1. Regarding Eq. 8. Since the \lambda is near 1 (for the approximation in Eq. 10), how do the terms of order 3 and beyond are neglected?

        Using the arguments of answer 3, and assuming that  are terms that contain factors of the form    , they are tiny and        , so the factor     .  It has no significant contribution, so the behavior of  is the same.

  1. I do not see the influence of the equation of the state parameters in the analysis of stability. Since the stability analysis is built from the efficiency it seems that the working fluid information is lost when the parameters of the model are fixed to obtain the \eta_C/2 efficiency. This should be clarified in the paper and in the concluding remarks.

Indeed, the parameters a and b do not appear in the stability expressions, but the efficiency implicitly has this dependence, after the approximation mentioned above, Ladino-Luna [36].

Reviewer 2 Report

The paper analyzes local and global stability of endoreversible engine. The authors claim they make use of van der Waal gas, but it enters in an approximate form. In particular, why authors take \lambda_vW = 1 should be clearly explained. What is the significance of this value being unity?

Secondly, I could not see the influence of two parameters a and b of vW gas on the stability equations. So, where exactly is it different from the ideal gas behavior studied in earlier references. Am I missing something here?

l.165 should be rewritten.

Eq. 23 may have dy/dt

extensive english language corrections are needed. I only point out a few,

such as in l.186, l.212.

Author Response

The manuscript was substantially modified according to the comments, so it is necessary to mention that the paragraphs do not follow the original order of the manuscript, but all the observations were made. 

1.- The paper analyzes local and global stability of endoreversible engine. The authors claim they make use of van der Waal gas, but it enters in an approximate form. In particular, why authors take \lambda_vW = 1 should be clearly explained. What is the significance of this value being unity?

Using the Gutkowicz-Krusin et al. formalism [35], Ladino-Luna [36]  obtained ec. (8). Nevertheless, b parameter in the van der Waals equation is negligible compared with the spanned volumes of the engine, namely   and . So we have  , and  is the compression ratio of the engine, which has a particular value depending on the chosen cycle: Diesel, Otto, etc. In the case of the Diesel cycle, ; and for the Otto cycle, . Thus, considering  and substituting it in (9)  we obtain ,  for the simplicity calculus, we use  and substitute it in (8) we obtain   This substitution, by observing the behavior of the plots   as functions of , with , leads to the behavior shown in figure 2. It is even shown an interval of , namely, , in which real plants usually work;  Bejan A., 1988, Advanced Engineering Thermodynamics (New York: Wiley) [38], Velasco S., Roco J.M.M., Medina A., White J.A., and Calvo-Hernández A., 2000, J. Phys. D: Appl. Phys. 33 355 [39]. The behavior of  and  is the same, so there is no loss of generality in this study.

2,- Secondly, I could not see the influence of two parameters a and b of vW gas on the stability equations. So, where exactly is it different from the ideal gas behavior studied in earlier references. Am I missing something here?

Indeed, the parameters a and b do not appear in the stability expressions, but the efficiency implicitly has this dependence, after the approximation mentioned above.

3.- l.165 should be rewritten.

Eq. 23 may have dy/dt

It was corrected.

Reviewer 3 Report

The authors presented the global stability analysis of the Curzon-Ahlborn engine with a working substance that satisfies the van der Waals equation state and they used Lyapunov stability theory when the heat engine is operating at maximum power regime. They analyzed the steady-state of the intermediate temperatures and its asymptotic behavior for the heat engine in terms of  thermodynamic viewpoint. I think this study can be published after minor revision.

1. The article should be thoroughly reviewed in terms of spelling and language structure.

2. The references should be updated with the recent studies related to finite-time thermodynamics published in last two years. 

3. The equation numbers should be revised. 

Author Response

The manuscript was substantially modified according to the comments, so it is necessary to mention that the paragraphs do not follow the original order of the manuscript, but all the observations were made. 

The authors presented the global stability analysis of the Curzon-Ahlborn engine with a working substance that satisfies the van der Waals equation state and they used Lyapunov stability theory when the heat engine is operating at maximum power regime. They analyzed the steady-state of the intermediate temperatures and its asymptotic behavior for the heat engine in terms of  thermodynamic viewpoint. I think this study can be published after minor revision.

  1. The article should be thoroughly reviewed in terms of spelling and language structure.

It was corrected

  1. The references should be updated with the recent studies related to finite-time thermodynamics published in last two years. 

Add References 2021 y 2022 [24-30]

  1. The equation numbers should be revised. 

It was corrected

Reviewer 4 Report

1.. Results are very brief and you need to discuss in more details and mechanism.

2.Conclusions section is written briefly. Please extend this part with more achievements and results.

3.The accuracy and validity of their scheme are required to significantly enhance the quality of their work.

4.Nomenclature is needed. Revise it carefully and align the text.

5.Some graphical results are not clear, provide better images so I can read it.

Author Response

Thank you for your comments on our paper, please see attached file.

Round 2

Reviewer 1 Report

I have read the revised version of the manuscript. The Authors have answered all my questions, and the modifications to the previous version are adequate. Thus, I recommend its publication in Entropy.

Reviewer 3 Report

It can be published in this form.

Reviewer 4 Report

The manuscript has been
sufficiently improved.